

# A comprehensive review of deep learning in EEG-based emotion recognition: classifications, trends, and practical implications

Weizhi Ma, Yujia Zheng, Tianhao Li, Zhengping Li, Ying Li and Lijun Wang

School of Information Science and Technology, North China University of Technology, Beijing, China

## ABSTRACT

Emotion recognition utilizing EEG signals has emerged as a pivotal component of human–computer interaction. In recent years, with the relentless advancement of deep learning techniques, using deep learning for analyzing EEG signals has assumed a prominent role in emotion recognition. Applying deep learning in the context of EEG-based emotion recognition carries profound practical implications. Although many model approaches and some review articles have scrutinized this domain, they have yet to undergo a comprehensive and precise classification and summarization process. The existing classifications are somewhat coarse, with insufficient attention given to the potential applications within this domain. Therefore, this article systematically classifies recent developments in EEG-based emotion recognition, providing researchers with a lucid understanding of this field's various trajectories and methodologies. Additionally, it elucidates why distinct directions necessitate distinct modeling approaches. In conclusion, this article synthesizes and dissects the practical significance of EEG signals in emotion recognition, emphasizing its promising avenues for future application.

## INTRODUCTION

Recently, there has been a growing interest in the computation and recognition of emotions (*Kamble & Sengupta, 2023*). Within the ever-evolving realm of human–computer interaction (*Moin et al., 2023*), emotion recognition based on EEG signals has emerged as a prominent and highly researched topic (*Wang et al., 2023*). Currently, the applications of emotion recognition based on EEG extend well beyond the boundaries of human–computer interfaces. It has found widespread utility in diverse domains, including the treatment of psychological disorders such as post-traumatic stress disorder (PTSD) (*Cruz et al., 2023*) and depression (*Miljevic et al., 2023*), as well as addressing issues like fatigue-induced driving incidents (*Li et al., 2023c*). Additionally, it has contributed to developing systems related to emotional intelligence, such as intelligent recommendation systems utilizing EEG signals (*Panda et al., 2023*) and immersive virtual reality experiences (*Choi et al.,*

Corresponding authors
Zhengping Li, lizp@ncut.edu.cn
Lijun Wang, ljwang@outlook.com

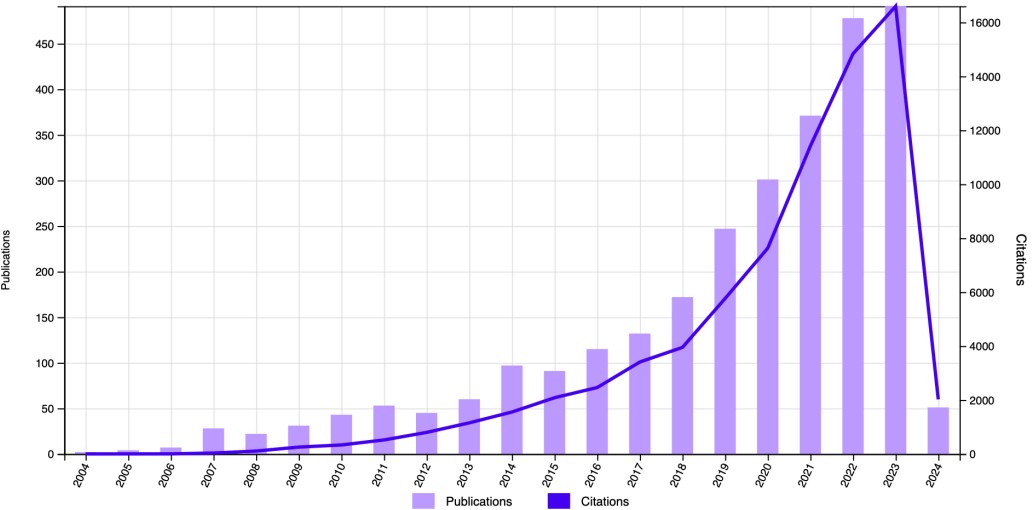

**Figure 1** **The publication and citation rate of articles on emotion recognition utilizing EEG signals continues to rise rapidly.** (Data Source: Web of Science Citation Report; Keyword: EEG, emotion recognition, emotional classification, and emotional computation; Date Range: January 1st, 2004, to March 18th, 2024; Database Coverage: All databases except preprints).

*2023*). Consequently, analyzing emotions through EEG signals holds immense practical importance and demonstrates its potential for multifaceted applications.

A diverse array of data sources is presently encompassed within the domain of emotion recognition, encompassing EEG signals, facial expressions (*Karnati et al., 2023*; *Adyapady & Annappa, 2023*; *Leong et al., 2023*), and speech signals (*Hammed & George, 2023*; *Anthony & Patil, 2023*; *Al-Dujaili & Ebrahimi-Moghadam, 2023*). It is crucial to recognize that facial and vocal expressions may be subject to conscious concealment or manipulation, compromising their objectivity and authenticity compared to EEG signals (*Parry et al., 2019*). In contrast, EEG signals are increasingly acknowledged for their inherent objectivity and authenticity in facilitating emotion recognition, prompting a growing inclination among researchers to prioritize them as the primary raw data source. Figure 1 illustrates that emotion recognition leveraging EEG signals has garnered considerable attention in recent years.

## Summary of previous reviews

*Roy et al. (2019)* underscores the potency of deep learning in acquiring enhanced feature representations directly from raw EEG data. The authors meticulously reviewed 154 deep learning-based EEG articles published from January 2010 to July 2018 to distill the prevailing trends and methodologies in this context. Their objective was to glean insights that could inform future research endeavors. Notable areas of exploration spanned epilepsy, sleep studies, brain-computer interfaces, cognitive investigations, and emotion detection. Remarkably, the analysis revealed a dominant presence of convolutional neural networks (CNNs) in 40% of the studies, recurrent neural networks (RNNs) featuring in 13% of

the cases, while a significant portion of these studies—almost half—focused on training models utilizing either raw or preprocessed EEG time series data.

*Suhaimi, Mountstephens & Teo (2020)* delves into the progress of EEG-based emotion recognition between 2016 and 2019. It highlights various factors, including the type and delivery of emotional stimuli, the sample size of studies, EEG hardware employed, and machine learning (ML) techniques utilized. The article introduces a novel approach involving virtual reality (VR) for presenting stimuli, drawing motivation from a review of VR research within the emotion recognition domain.

*Rahman et al. (2021)* conducts a comprehensive review of published research utilizing EEG signal data to explore potential links between emotions and brain activity. The article elucidates the theoretical underpinnings of basic emotions and outlines prominent techniques for feature extraction, selection, and classification. A comparative analysis of study outcomes is provided, alongside a discussion of future directions and significant challenges anticipated in the development.

*Li et al. (2022d)* offers a comprehensive analysis from a researcher's perspective. This review delves into the intricacies inherent to the science of the field, examining the psychological and physiological foundations, elucidating distinct conceptual pathways and theoretical frameworks, elucidating the driving forces behind such investigations, and rationalizing the research and application of these methodologies.

*Dadebayev, Goh & Tan (2022)* offers a critical assessment of consumer-grade EEG devices and contrasts them with their research-grade counterparts. They highlight critical aspects of EEG-based emotion recognition research and identify primary challenges in system development, focusing on data collection and machine learning algorithm performance with commercial EEG devices.

*Jafari et al. (2023)* focus lies in utilizing deep learning (DL) methodologies to recognize emotions from EEG signals. The article extensively engages with the pertinent academic literature, engaging in an in-depth examination of the complexities inherent in EEG-based emotion recognition. It emphasizes the promise held by DL techniques to address and ameliorate these challenges. Additionally, the article outlines potential directions for future studies in utilizing DL techniques for recognizing emotions. A particularly auspicious trajectory for future investigations involves the exploration of system-on-chip (SoC) architectures leveraging field-programmable gate arrays (FPGAs) and application-specific integrated circuits (ASICs) for enhancing emotion recognition from EEG signals. Finally, in its concluding segment, the article summarizes its primary findings, affirming the emergence of DL techniques as promising solutions for EEG-based emotion recognition. It also presents a comprehensive review of studies employing DL methodologies.

*Khare et al. (2023)* extensively examines emotion recognition techniques, emphasizing the utilization of diverse information sources such as questionnaires, physical cues, and physiological signals, including EEG, ECG, galvanic skin response, and eye tracking. The review encompasses various physical cues like speech and facial expressions, alongside a comprehensive survey of emotion models and stimuli used to evoke emotions. The article systematically analyzes 142 journal articles following PRISMA guidelines, offering insights into existing research, available datasets, and potential challenges. Challenges

include variable signal lengths in emotion recognition studies due to diversity in system specifications, hindering trust in model decisions. Existing automated systems face trust issues due to discrepancies with prior knowledge and limited real-time support, necessitating transparent model explanations. Future research directions suggest leveraging federated meta-learning to create accurate and versatile models across diverse datasets for specific applications.

In recent years, the continuous evolution of deep learning has led to the emergence of an increasing number of models designed for EEG-based classification. Consequently, classification accuracy has steadily improved over time. While some researchers have generalized and summarized these methods, they have typically approached the topic from diverse technical perspectives (_Prabowo et al., 2023_; _Vempati & Sharma, 2023_). These approaches provide valuable insights into the theoretical underpinnings and research motivations. Table 1 summarizes the reviews mentioned above for reference succinctly. However, there remains a gap in the literature regarding the comprehensive categorization and classification of this field's diverse directions and methodologies. This gap poses a challenge for newcomers to the field who seek a quick understanding of its developmental direction. Hile's previous researchers have attempted to classify methods within emotion recognition based on EEG signals. However, these attempts have often needed more clarity in perspective and have not sufficiently explained the rationale behind their classification.

To tackle the issues above, this article introduces a novel classification approach aimed at effectively categorizing existing studies in the domain of EEG-based emotion recognition. Given the intrinsic variability in EEG signals across individuals, this categorization method delineates two specific directions: subject-independent and subject-dependent. These directions correspond to research involving classification across individuals and research that does not cross individual boundaries.

We can effectively synthesize the underlying research principles and motivations by categorizing ongoing research into these two domains. Within the subject-independent direction, researchers aim to enhance model generalization and robustness. Their primary objective is to maximize the extraction of standard essential information, mainly when dealing with distinct source and target domains. This direction significantly emphasizes the model's ability to generalize effectively and maintain robustness across differing domains. Conversely, in the subject-dependent direction, researchers are dedicated to improving the model's feature extraction capabilities to enhance classification accuracy. The aim is to extract and incorporate a comprehensive array of sample features, thus elevating classification accuracy. Moreover, this article comprehensively overviews prevailing methods and models applied in diverse domains. Finally, it offers a cross-disciplinary perspective on the field's practical relevance and potential future applications.

## Contribution of the review

The primary contributions of this article are as follows:

**Table 1** Summary table of review articles.

| Year | References | Selecting research article years | Main contributions |
|---|---|---|---|
| 2019 | *Roy et al. (2019)* | 2010–2018 | The article examines DL applications to EEG data, outlining trends in the DL-EEG field, emphasizing origins, principles, data sources, EEG processing, DL techniques reported findings, and reproducibility levels. |
| 2020 | *Suhaimi, Mountstephens & Teo (2020)* | 2016–2019 | The article analyzes emotion classification studies introducing novel EEG-based approaches. It explores methods using VR for emotional stimuli presentation and underscores the need for new VR-based databases. |
| 2021 | *Rahman et al. (2021)* | 2009–2021 | The article suggests enhancing emotion estimation from EEG signals with a hybrid artifact removal techniques, emphasizing the potential of deep learning methods like CNN, DBN, and RNN to distinguish emotional states and enhance accuracy in future research |
| 2022 | *Li et al. (2022d)* | 2016–2022 | The article succinctly reviews EEG-based emotion recognition research, covering advancements, fundamental principles, mainstream, and state-of-the-art technology lines, and standard evaluation methods. |
| 2022 | *Dadebayev, Goh & Tan (2022)* | 2015–2020 | The article examines ten studies employing popular consumer-grade EEG devices, analyzing participant numbers, stimulus types, extracted features, and utilized machine learning algorithms. |
| 2023 | *Jafari et al. (2023)* | 2016–2023 | The article examines emotion classification methods, underscoring EEG signal importance and the superiority of DL over ML for emotion recognition. |
| 2023 | *Khare et al. (2023)* | 2014–2023 | The article examines human emotions, highlighting how decomposition techniques aid in feature extraction from physiological signals for emotional recognition systems. It also stresses the importance of features and data fusion in improving system performance. |
| 2023 | *Prabowo et al. (2023)* | 2017–2023 | The article analyzes trends in human emotion recognition via EEG signals, with a focus on datasets, classifiers, and research contributions. |
| 2023 | *Vempati & Sharma (2023)* | 2016–2022 | The article surveys EEG databases, preprocessing methods, feature extraction, and selection techniques, along with analyzing emotion classification methods based on AI, machine learning, and deep learning. |

- Novel classification framework: This article introduces a well-structured classification method based on research directions, encompassing subject-independent and subject-dependent approaches. It provides a clear rationale for classification and elucidates the distinct research objectives within each direction.
- Comprehensive method summaries: Given the inherent variability in research objectives across diverse directions, this article presents a thorough synthesis of scholarly works spanning the field's six years from 2018 to 2023. Emphasizing distinct modeling approaches, it furnishes a comprehensive overview of methodologies pertinent to each research direction.
- Detailed dataset comparison: This article explores both mainstream and state-of-the-art datasets, offering a comparative analysis of these four prominent datasets.
- Comprehensive overview of EEG signal-based classification: This article provides a comprehensive summary of the recent research developments in EEG-based classification. It further delves into the practical significance and potential applications of this field. In addition, the article suggests several methods that can be integrated with other domains to enhance the value and utility of EEG-based classification.

## Target audience of the review

This review is dedicated to introducing innovative categorization methods for emotion classification using EEG signals. It is intended for a diverse audience, including professionals from various domains such as human–computer interaction, brain-computer interfaces, emotion recognition, affective computing, and EEG studies. We recommend this review to the following readers:

- Researchers and scholars are interested in gaining insights into publicly available datasets for emotion recognition through EEG signals. This review provides an overview of these datasets, highlighting their commonalities and distinctions.
- Researchers and scholars are seeking to comprehend the diverse research directions in emotion recognition *via* EEG signals. It delves into the core concepts of different research avenues, allowing readers to grasp their distinct approaches and objectives.
- Researchers and scholars are eager to explore the future trajectory of emotion recognition using EEG signals and its potential integration with other related fields. This review offers valuable perspectives on the evolving domain's prospective developments and interdisciplinary connections.

## Organization of the review

The article is organized as follows: In 'Survey methodology', we expound on our meticulous approach to the impartial and comprehensive selection and evaluation of pertinent literature. We provide insights into the classification, assessment, and interrelation of analytical studies, all undertaken by the taxonomy proposed within this article. 'Task and datasets' is dedicated to elucidating the principal tasks within EEG-based sentiment analysis. It also extensively analyzes mainstream and cutting-edge datasets, delving into their shared characteristics and distinctions. 'EEG Signal processing' outlines the preliminary steps preceding the design of EEG signal-based models for sentiment analysis, emphasizing preprocessing and feature extraction procedures applied to raw EEG signals. 'Machine learning approaches' delves into the nascent stages of the field, wherein traditional machine learning models were initially employed for classification development. Within 'Classification and methods', we outline the classifications introduced in this study. Furthermore, we provide succinct summaries and insightful analyses for each classification and an overview of the pertinent literature. 'Significance and applications' explores the field's relevance, shedding light on prospective integration opportunities with other domains. We proffer suggestions for the fusion of this field with complementary research areas. In 'Discussion', we discussed some of the field's current challenges and future directions. 'Conclusion' offers a comprehensive summary of the entire text, meticulously scrutinizing the identified challenges and charting a path forward for future investigations.

## SURVEY METHODOLOGY

This review systematically compiles and categorizes the latest research models in EEG-based emotion recognition published over the last six years, starting in 2018. The scope of these models encompasses contributions from English journals, conference articles,

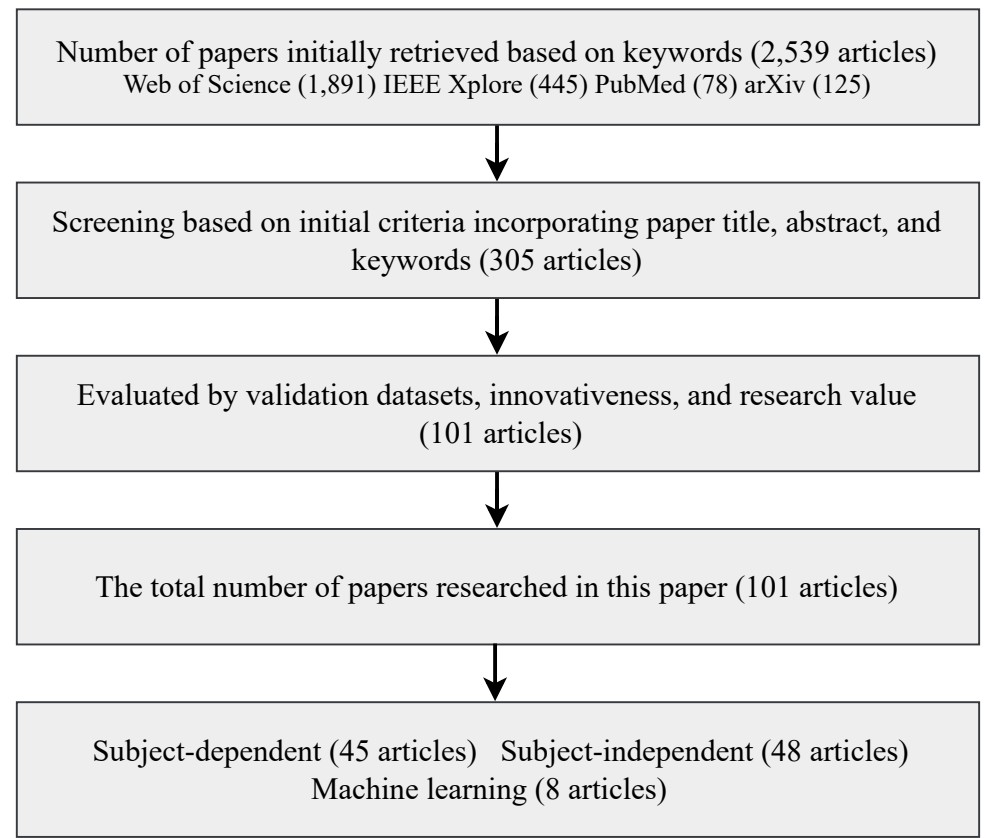

**Figure 2   Flowchart of the article screening process.**

and electronic preprints. In our quest for comprehensive coverage, we sourced and organized these articles from reputable platforms, including Web of Science (https://www. webofscience.com), arXiv (https://arxiv.org/), IEEE Xplore (https://ieeexplore.ieee.org), and PubMed (https://pubmed.ncbi.nlm.nih.gov/). The keywords guiding our article selection process encompassed EEG, deep learning, emotion recognition, emotional classification, and emotional computation. In the past six years, from 2018 to 2023, 2,539 articles were initially retrieved. Following rigorous screening, 101 articles were ultimately selected for detailed analysis, as depicted in Fig. 2.

In order to rigorously uphold the principles of impartiality, objectivity, and practicality, a meticulous manual screening process was implemented for the collected articles. This screening process was structured into two pivotal stages.

The screened models and articles underwent comprehensive review and analysis during the initial stage. This review encompassed an initial assessment of the articles, delving into the foundational datasets upon which the article and model were predicated. Articles selected for inclusion in this study were based on mainstream publicly available datasets, as elucidated within this manuscript. Self-collected datasets and non-mainstream datasets were omitted from consideration. The analysis involved scrutinizing the articles identified

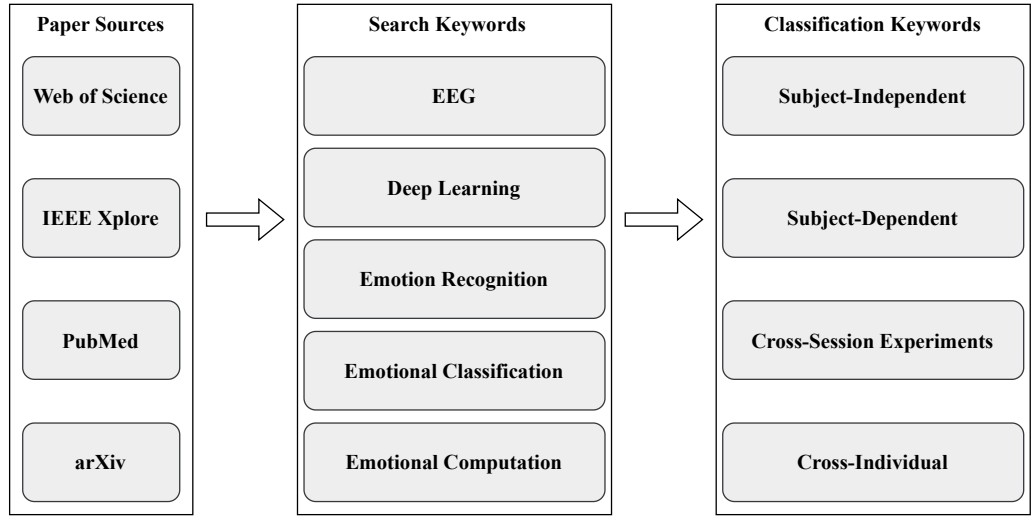

**Figure 3** **Flowchart of survey methodology.**

in the initial review to discern those possessing high research value and innovation. This evaluative process was centered on three critical criteria:

- The extent of contribution to the field, including whether the article filled a lacuna or proposed a novel solution to an existing problem.
- The originality of the methodology and technology. The proposed methodology should either be novel or offer enhancements and expansions to existing methodologies, addressing their deficiencies with innovative approaches.
- The efficacy of the model. The proposed model should demonstrate superior performance compared to contemporaneous models or exhibit improved effectiveness over its predecessors in the case of modifications to existing models.

Subsequently, in the second stage, the collected literature was meticulously categorized according to the classification framework outlined in this review. This classification considered factors such as the presence of subject-independent or subject-related experiments and cross-session or cross-individual experiments. The detailed process is depicted in Fig. 3.

# TASK AND DATASETS

This section primarily delves into the prominent tasks associated with the computation of emotions using EEG signals. It provides a concise introduction to the publicly accessible datasets currently widely employed in this domain.

## Classification tasks

Emotion recognition based on EEG signals encompasses many tasks, including data acquisition, preprocessing, feature extraction, and classification model design. This article predominantly focuses on classification model design, the prevailing pursuit in current

research. Prior studies have made significant advancements in preprocessing raw data and extracting relevant features. Consequently, recent research efforts have shifted their focus toward the design of classification models.

The domain of classification model design involves leveraging the principles and models of deep learning to learn from preprocessed EEG signals. This enables the model to analyze signals, effectively enhancing emotion recognition accuracy. Furthermore, it empowers the model to predict emotions in unlabeled EEG signals, facilitating its application in downstream tasks.

## Datasets

In the realm of EEG emotion recognition, three publicly accessible mainstream datasets currently stand out: DEAP (*Koelstra et al., 2011*), SEED (*Duan, Zhu & Lu, 2013*; *Zheng & Lu, 2015*), and SEED-IV (*Zheng et al., 2018*). These datasets are widely recognized and frequently employed by researchers in EEG emotion recognition for classification tasks. As the field of human–computer interaction continues to advance, more researchers are engaging in emotion recognition based on EEG signals, leading to new datasets like DENS (*Asif et al., 2023*). The subsequent section provides a brief overview of these commonly used datasets.

The DEAP dataset comprises data collected from 32 subjects, evenly split between 16 males and 16 females. Each subject participated in a single experiment, viewing 40 music videos. The experimental procedure for each subject included a 3-second baseline recording, followed by viewing a 60-second music video. Afterward, the subjects spent 15 seconds completing the Self-assessment Manikins (SAM) scale, which includes four dimensions: valence, arousal, dominance, and liking. Subsequently, the subjects watched another 60-second music video.

The SEED and SEED-IV datasets were provided by the BCMI laboratory at Shanghai Jiao Tong University and included data from 15 subjects, comprising seven males and eight females. The subjects have an average age of 23.27 years, with a standard deviation of 2.37 years. The experimental design for these datasets involves each subject participating in three separate trials spaced one week apart. In each trial for every subject, there is an initial 5-second hint to signal the commencement of the trial, followed by a 4-minute movie clip, a 45-second self-assessment period, and concluding with a 15-second rest period.

The DENS dataset encompasses data from 40 participants with an average age of 23.3 years and a standard deviation of 1.25 years. Each participant was subjected to a single experiment, which involved viewing 11 music videos. These videos included nine emotional stimuli and two non-emotional stimuli. The experimental protocol for each participant followed a consistent sequence: an initial 80-second baseline recording, followed by viewing a 60-second video segment. Subsequently, participants completed the Self-assessment Manikins (SAM) scale, which measured various emotional dimensions, including valence, arousal, dominance, liking, familiarity, and relevance. Finally, participants proceeded to watch another 60-second video segment. Table 2 compares the commonalities and differences between these four datasets.

**Table 2  Similarities and differences between the four EEG datasets.**

| Item | DEAP | SEED | SEED-IV | DENS |
|---|---|---|---|---|
| Subjects | 32 | 15 | 15 | 40 |
| Video clips number | 40 | 15 | 24 | 11 |
| Video clips duration | 1-min | 4-min | 2-min | 1-min |
| Number of EEG channels | 32 | 62 | 62 | 128 |
| Sampling rate | 128 Hz | 200 Hz | 200 Hz | 250 Hz |

**Notes.**

Note: This table compares four datasets: DEAP, SEED, SEED-IV, and DENS. "Subjects" denotes the total number of participants in the experiment; "Video clips number" denotes the number of videos each participant watches; "Video clips duration" denotes the length of each video; "Number of EEG channels" denotes the number of electrodes on the EEG cap used in each dataset; and "Sampling rate" denotes the sampling rate of the data in each dataset.

## EEG SIGNAL PROCESSING

Raw EEG signals exhibit high temporal resolution and harbor pertinent information. However, they are also susceptible to noise and artifacts, necessitating preprocessing as an indispensable and pivotal precursor to signal analysis. The intricacy of EEG signals encompasses temporal, spatial, and frequency domain characteristics, thereby rendering the extraction of pertinent features an imperative facet in model design. Consequently, this chapter explores the intricacies of EEG signal preprocessing and feature extraction, exploring methodologies to optimize signal fidelity and extract salient features across temporal, spatial, and frequency domains.

### Preprocessing

During EEG data collection, it is imperative to acknowledge the presence of various forms of noise and artifacts stemming from intrinsic subject-related factors, experimental environmental conditions, or equipment-related influences. Consequently, raw data collected from EEG recordings cannot be directly utilized for analysis and classification purposes. Noteworthy among these disturbances are electrooculographic, electrocardiographic, electromyographic, and industrial frequency artifacts. Electrooculographic artifacts typically manifest below 4 Hz, attributed to eye movements or blinking. Electrocardiographic artifacts, occurring at approximately 1 Hz, result from cardiac contractions. Electromyographic artifacts, characterized by frequencies exceeding 25 Hz, stem from muscular activities, particularly in temporal, prefrontal, and cervical regions. Industrial frequency interference, typically at 50 or 60 Hz, originates from utility sources. Hence, mitigating these noise and artifacts is imperative for accurate analysis and classification of EEG signals, facilitating extracting meaningful insights from the recorded data. Consequently, removing such disturbances is essential to ensure the integrity and reliability of EEG signal analysis and classification endeavors.

Given the criticality of mitigating noise and artifacts in EEG signals, numerous studies focus on preprocessing techniques. Commonly employed methods for artifact removal encompass regression methods, wavelet transform, filtering, principal component analysis (PCA), and independent component analysis (ICA). The regression method (*Klados et al., 2011*), a traditional approach in EEG preprocessing, is versatile, operating effectively

in both the time and frequency domains (*Whitton, Lue & Moldofsky, 1978*). Wavelet transformation (*Kumar et al., 2008*) facilitates the conversion of time-domain information into both temporal and frequency-domain representations, enabling artifact removal through subsequent thresholding techniques. Various filtering methodologies, such as adaptive filtering (*Correa et al., 2007*), Wiener filtering (*Somers, Francart & Bertrand, 2018*), and high-pass filtering (*Winkler et al., 2015*), offer additional avenues for noise reduction during preprocessing. PCA (*Berg & Scherg, 1991*) relies on eigenvalues derived from the covariance matrix, providing a powerful means to extract meaningful features while attenuating artifacts. ICA (*Jung et al., 1998*), an extension of PCA, further enhances artifact removal capabilities by decomposing EEG signals into statistically independent components. Collectively, these advanced techniques signify significant advancements in EEG preprocessing, furnishing researchers with many robust tools to remove noise and artifacts from raw EEG signals effectively.

### Feature extraction

Initially, EEG research primarily focused on the time domain of EEG signals, resulting in the extraction of significant features but yielding suboptimal classification outcomes. However, with the evolution of research methodologies, there has been a shift towards acknowledging and incorporating spatial and frequency domain information and exploring joint information integration.

In the time domain, methods such as the Higher-Order Crossing (HOC) method (*Petrantonakis & Hadjileontiadis, 2009*) and Hjorth features (*Patil, Deshmukh & Panat, 2016*) have garnered widespread adoption due to their efficacy. Regardless of the number of channels utilized for EEG signal acquisition, be it 32, 62, or 128, it is evident that there exists a spatial relationship among these channels, reflective of the underlying neural architecture. Consequently, this spatial relationship must be noticed during feature extraction. In spatial domain processing, common spatial patterns (CSP) (*Wang, 2011*) and hierarchical discriminant component analysis (HDCA) (*Alpert et al., 2013*) techniques are commonly employed to capture spatial dependencies effectively. The frequency domain information, recognized as pivotal in characterizing EEG signals, has garnered increasing attention from researchers. Utilizing techniques like the Fast Fourier Transform (FFT) (*Murugappan & Murugappan, 2013*) to segment EEG signals into distinct frequency bands $(\delta, \theta, \alpha, \beta, \gamma)$, researchers then analyze these segments using power spectral density (PSD) (*Zheng & Lu, 2015*; *Wang, Nie & Lu, 2014*) or differential entropy (DE) (*Duan, Zhu & Lu, 2013*) to extract frequency domain features. By incorporating spatial and frequency domain information alongside traditional time-domain analyses, researchers can gain a more comprehensive understanding of EEG signals, thereby enhancing the efficacy of feature extraction and classification methodologies.

## MACHINE LEARNING APPROACHES

In EEG-based emotion recognition, initial classification methodologies included traditional machine learning techniques like support vector machine (SVM), K-nearest neighbor (KNN), random forest (RF), and decision tree (DT). These conventional machine learning

approaches laid crucial groundwork for developing contemporary sentiment classification models.

**Support vector machine (SVM):** *Li, Song & Hu (2018)* employs SVM to assess emotion recognition performance in the DEAP and SEED datasets. Furthermore, the study explores the importance of different EEG features for cross-subject analysis in the SEED dataset. It employs manual feature selection across various channels, brain regions, and other relevant factors. *Asghar et al. (2019)* presents a 2D spectrogram to retain spectral and temporal components before feature extraction. Raw features are then extracted using a pre-trained AlexNet model, and emotional states in SEED and DEAP datasets are classified using SVM and KNN algorithms. *Wang, Hu & Song (2019)* slices EEG signals into fixed-length segments using a sliding window and computed EEG spectrograms with short-time Fourier transform. Channel reduction was performed based on normalized mutual information (NMI) and thresholding from the inter-channel connectivity matrix. Emotion recognition in the DEAP dataset utilized SVM.

**K-nearest neighbor (KNN):** *Mert & Akan (2018)* examines the use of empirical mode decomposition (EMD) and its multivariate expansion (MEMD) for emotion recognition. It proposes a MEMD-based feature extraction method for multichannel EEG signals and validates it on the DEAP dataset using KNN. *Li et al. (2018b)* investigates how EEG signals from different frequency bands and channel configurations impact emotion recognition accuracy. Emotion recognition uses a KNN model. *Liu et al. (2018)* presents a hybrid feature extraction method for EEG emotion recognition, combining empirical pattern decomposition domains with optimal feature selection through sequence inverse selection. Classification is performed using KNN and SVM on the DEAP dataset.

**Random forest (RF) or decision tree (DT):** *Gupta, Chopda & Pachori (2018)* introduces an efficient emotion recognition method utilizing the Flexible Analytic Wavelet Transform (FAWT). Features are extracted from decomposed sub-band signals of EEG signals using information potential. Smoothed feature values are input into RF and SVM classifiers. *Jiang et al. (2019)* introduces a DT Classifier with Sequential Backward Selection (DT-SBS) and assesses cross-subject performance on the DEAP dataset.

Employing traditional machine learning methodologies in sentiment categorization has laid a fundamental framework and facilitated the progression toward developing more intricate models. Techniques such as SVM, KNN, DT, and RF have served as cornerstone methodologies, offering a robust basis for future research endeavors and advancements.

## CLASSIFICATION AND METHODS

In previous research endeavors, the classification of EEG-based emotion recognition can be broadly categorized into two principal directions: subject-independent and subject-dependent, the former encompassing cross-individual classification and the latter within non-cross-individual classification. This classification approach serves two primary purposes: first, it aids in the systematic organization and comprehensive understanding of the existing models and studies in the EEG emotion recognition domain. It offers a rational framework for categorizing and classifying these diverse studies, facilitating

clarity and coherence within the field. Moreover, this classification approach holds practical significance due to the inherent characteristics of EEG signals, characterized by substantial individual variability. Consequently, the research on EEG signals necessitates a dichotomous approach: subject-independent and subject-dependent, each with distinctive objectives and applications.

Within the subject-dependent paradigm, each individual is independently trained, resulting in the development of personalized classification models. These models excel at capturing and deciphering the unique emotional characteristics of each individual. As a result, they enable highly personalized emotion recognition. This level of personalization proves invaluable in mental health, enabling accurate detection and diagnosis of an individual's emotional state. This, in turn, paves the way for precise and individualized interventions and treatments for mental health conditions.

Conversely, in the subject-independent domain, the focus is on enhancing the model's generalization capability through cross-individual training. These models are adept at adapting to a wide range of individuals, making them particularly well-suited for applications in large-scale scenarios involving sentiment analysis. For instance, they are instrumental in large-scale recommender systems, where creating personalized models for each individual is impractical. Cross-individual trained models prove more effective for sentiment recognition in such expansive contexts.

The subject-independent direction entails classifying emotions across individuals, while the subject-dependent approach does not cross these individual boundaries. In subject-dependent research, the dataset is typically partitioned to randomly designate a subset of samples for the training set. Conversely, within the subject-independent direction, all samples from a specific group of individuals are included in the training set. Consequently, these two distinct directions entail divergent foci and objectives, resulting in the utilization of varying methodologies. The subject-independent direction concentrates on enhancing the generalization and robustness of the model, emphasizing the extraction of standard critical information. This is particularly pertinent when addressing discrepancies between the Source Domain and Target Domain, making the model more adaptable to different data sources.

In contrast, the subject-dependent direction prioritizes improving the model's feature extraction capabilities for sample-specific data to enhance classification accuracy. Here, the aim is to facilitate extracting many sample-specific features, consequently elevating classification accuracy. This distinct set of objectives necessitates different methodological approaches in the two directions.

## Subject-dependent

The predominance of research on the subject-dependent approach in EEG-based emotion recognition can be attributed to the profound individual variability intrinsic to EEG signals. This research direction has garnered extensive attention, primarily due to the challenges associated with subject-independent classification, its significant practical importance, and real-world applications. As a result, many researchers have devoted their efforts to the subject-dependent paradigm. Research within this direction can be delineated into two

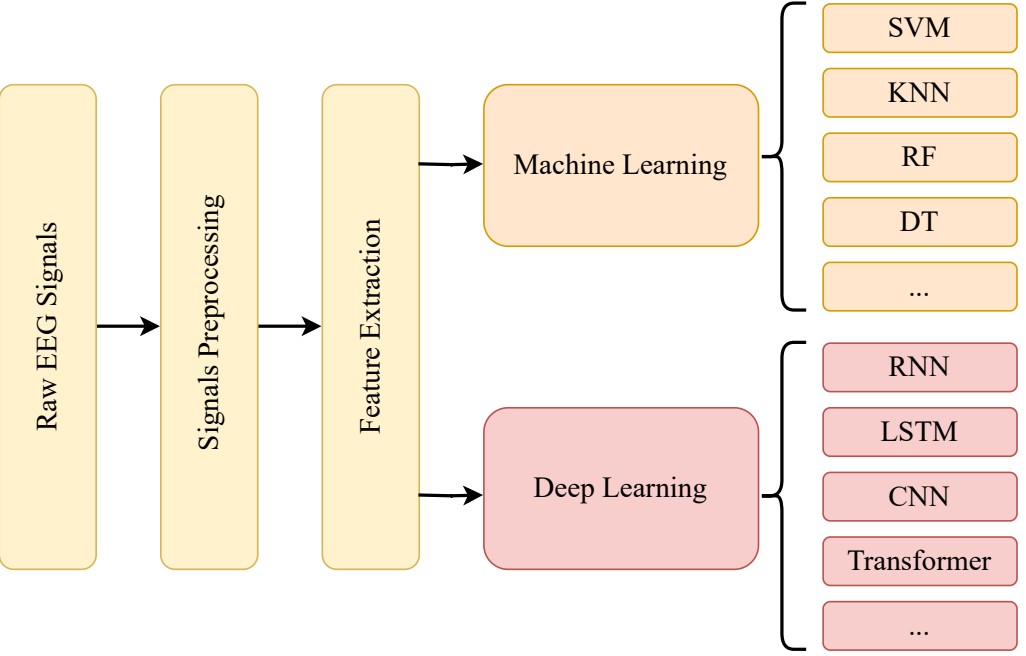

**Figure 4  Subject-dependent technology roadmap.**

primary phases: the initial phase primarily focused on classification using exclusively time-domain features. Subsequently, the second phase emerged, embracing a more comprehensive feature extraction methodology encompassing the time, spatial, and frequency domains. The technical diagram for this direction is visually represented in Fig. 4.

EEG signals, notable for their high temporal resolution characteristics, were initially introduced into the domain of natural language processing (NLP) through their integration into deep learning models. Initially, sequence data processing in deep learning predominantly relied on recurrent neural network (RNN); however, RNN often grapples with challenges such as vanishing or exploding gradients, particularly when confronted with lengthy sequences. Consequently, long short-term memory (LSTM) networks emerged as a solution for effectively processing prolonged sequence data. Applying deep learning techniques to EEG information commenced with classification tasks employing RNN and LSTM networks.

As the EEG field progressed, many scholars explored the spatial and frequency domain characteristics inherent in EEG signals. In parallel with the evolution of deep learning methodologies, CNN was introduced and gradually integrated into the domain of NLP. Similarly, CNN found application in EEG signal processing for tasks such as emotion recognition. CNN architectures excel in capturing local and hierarchical features within data, rendering them highly adept at processing EEG signals. The emergence of the Transformer architecture marked another significant advancement. Characterized by its self-attention mechanism capable of capturing long-range dependencies within sequences,

the Transformer architecture shifted the paradigm. The attention mechanism embedded within Transformer networks facilitates an enhanced focus on EEG features. Consequently, many networks leveraging Transformer architectures for EEG emotion recognition have surfaced in recent years.

### Recurrent neural network or long short-term memory

*Garg et al. (2019)* introduces a merged LSTM model for binary emotion recognition. It utilizes signal processing techniques, including wavelet transform and statistical measures for feature extraction and dimensionality reduction. *Li et al. (2019b)* presents R2G-STNN, a novel EEG emotion recognition method. It integrates spatial and time neural network architectures, employing a hierarchical feature learning mechanism to distinguish discriminative spatiotemporal EEG features. *Xing et al. (2019)* suggests employing a stacked autoencoder (SAE) to construct and solve linear EEG hybrid models, then utilizing LSTM-RNN models for classification. *Acharya et al. (2020)* uses EEG signals to classify negative emotions *via* emotion recognition, employing an LSTM network. Its main goal is to evaluate the LSTM model's emotion recognition performance while assessing human behavior across different age groups and genders. *Du et al. (2020)* introduces the Attention-based LSTM with Domain Discriminator (ATDD-LSTM) model, which utilizes LSTM for emotion recognition. This model captures the nonlinear relationship among EEG signals from diverse electrodes. *Prawira et al. (2021)* suggests using the Fast Fourier Transform (FFT) as a preprocessing filter to categorize emotions prior to employing RNN. *Abgeena & Garg (2023)* proposes a Stacked Long Short-Term Memory and Attention (S-LSTM-ATT) model.

### Convolutional neural network

*Yang et al. (2018)* leverage hybrid neural networks, merging CNN and RNN architectures. This fusion enables the classification of emotional states by extracting spatio-temporal representations from raw EEG streams. *Yang, Han & Min (2019)* presents a multi-column CNN-based model. *Hwang et al. (2020)* suggests an emotion recognition method utilizing CNN that aims to preserve local information. *Chao & Dong (2020)* suggests time-domain processing of EEG signals from each channel, aggregating features into a 3D matrix based on electrode locations. This matrix employs an advanced CNN with univariate and multivariate convolutional layers for emotion recognition. *Wang et al. (2020)* introduces Electric Frequency Distribution Maps (EFDMs) *via* Short Time Fourier Transform (STFT). It proposes a CNN framework utilizing residual blocks for automated feature extraction and emotion recognition *via* EFDMs. *Li et al. (2021a)* introduces a new framework, 3DFR-DFCN, which merges 3D feature representation with a dilated fully convolutional network (DFCN) for emotion recognition. *Song et al. (2021)* introduces a novel approach called graph-embedded convolutional neural network (GECNN), which combines local CNN features with global functional features to offer complementary sentiment information. *Li et al. (2021b)* applies Fast Fourier Transform (FFT) and Continuous Wavelet Transform (CWT) to extract EEG signal features from the DEAP dataset. It then constructs two CNN models for emotion recognition. *Xiao et al. (2022)* transformed the original signal into four-dimensional representations, encapsulating temporal, spatial, and frequency

domains, and presented a dedicated four-dimensional attention neural network (4D-aNN). *Li et al. (2022a)* introduces an efficient CNN and contrast learning (ECNN-C) method. It employs a novel convolutional block instead of the standard convolution to mitigate the computational burden of the model. *Sun et al. (2022a)* suggests a multi-channel model utilizing a parallel converter and a three-dimensional CNN (3D-CNN). *Bao et al. (2022)* put forth the MDGCN-SRCNN network, amalgamating graph convolutional network and convolutional neural network techniques to improve the comprehensive extraction of information across all three dimensions.

*Zhong et al. (2023a)* proposed a novel approach that combines the Tunable Q-factor Wavelet Transform (TQWT) with a Hybrid Convolutional RNN (HCRNN) to enhance information extraction across the temporal, spatial, and frequency domains. *Li et al. (2023b)* extracts time, space, and connectivity features from EEG signals surrounding the head. These features are utilized for emotion recognition *via* the proposed spatial–temporal-connectivity multiscale convolutional neural network (STC-CNN) model. *Ramzan & Dawn (2023)* proposes that fusing deep learning models such as CNN and LSTM-RNN leads to improved performance in analyzing emotions using EEG signals. *Iyer et al. (2023)* introduces a hybrid model for emotion detection based on CNN and LSTM. *Aldawsari, Al-Ahmadi & Muhammad (2023)* employs a lightweight deep learning method, specifically a one-dimensional convolutional neural network (1D-CNN), for analyzing EEG signals and classifying emotional states. *Yuvaraj et al. (2023)* leverages a pre-trained 3D-CNN MobileNet model for transfer learning, extracting space–time representations of EEG signals to obtain emotion recognition features. *Dondup, Manikandan & Cenkeramaddi (2023)* proposes EEG-based emotion recognition utilizing variational mode decomposition (VMD) and CNN.

### Transformer

*Arjun, Rajpoot & Panicker (2021)* presents two visual transformer-based methods for the emotion recognition of EEG signals. One method involves generating two-dimensional images using continuous wavelet transform (CWT), while the other operates directly on the original signals. *Wang et al. (2021b)* introduced the Joint-Dimension-Aware Transformer (JDAT) model, which concurrently attends to information in the time domain, spatial domain, and frequency domain, leading to marked improvements in performance. *Aadam et al. (2022)* employs a novel transformer-based architecture called Perceiver for EEG emotion recognition. *Li et al. (2022b)* proposes an automated Transformer Neural Architecture Search (TNAS) framework using a multi-objective evolutionary algorithm (MOEA). *Sun et al. (2022b)* proposes DBGC-ATFFNet-AFTL, an adaptive converter feature fusion network with adapter tuning migration learning based on two-branch dynamic graph convolution. *Liu, Zhou & Zhang (2022)* introduces a Temporal and Channel Converter (TcT) model for emotion recognition, directly applicable to raw preprocessed EEG data. *Song et al. (2022)* introduces EEG Conformer, a compact convolutional transformer designed to incorporate local and global features within a unified EEG classification framework. *Guo et al. (2022)* examines the impact of individual

EEG channels on emotion recognition and introduces DCoT, a novel neural network model combining deep convolution and Transformer encoder.

*Xu et al. (2023b)* leveraged a multidimensional global attention mechanism to harness the complementarity of frequency, space, and time features within EEG single and developed a method that integrates CNN and Transformers, with a focus on capturing space and frequency domain details while accentuating time domain characteristics. *Lu, Tan & Ma (2023)* presented a technique in which EEG signals are fed into a Vision Transformer (VIT) model, facilitating the extraction of spatial, frequency, and temporal domain characteristics as a unified entity. *Qian et al. (2023)* presents AGCN-SAT, an adaptive graph convolution network with spatial attention and transformer. It utilizes an adaptive learning adjacency matrix to improve the network's capacity to capture local spatial features. *Sun et al. (2023)* proposes a multi-domain EEG feature-based emotion recognition transformer network named MEEG-Transformer. *Cao et al. (2023)* introduces BiCCT, an emotion recognition model that combines the bi-hemispheric asymmetry theory with the Compact Convolutional Transformer (CCT), achieving high performance with fewer training parameters. *Wan et al. (2023)* presents EEGformer, a transformer-based model for unified EEG analysis, integrating 1DCNN for automatic EEG channel feature extraction. *Zhong et al. (2023b)* proposes Bi-AAN, merging transformer architecture with the brain's asymmetric emotional response properties to model attention differences and mine long-term dependencies in EEG sequences. *Wei et al. (2023)* introduces TC-Net, comprising an EEG transformer module for feature extraction and an emotion capsule module for feature refinement and emotional state classification. *Cai et al. (2023)* proposes AITST, an emotion-related spatiotemporal transformer, to address the influence of different emotional states on person identification using practical EEG. *Lu, Ma & Tan (2023)* introduces CIT-EmotionNet, a novel CNN interactive transformer network, effectively integrating global and local features of EEG signals.

The trajectory of research endeavors over the past six years, as illustrated in Table 3, highlights a discernible trend: the evolving landscape of deep learning methodologies within EEG-based emotion recognition. A clear progression emerges, mirroring the continuous advancements in deep learning techniques. Beginning with earlier approaches such as RNN and LSTM, the trajectory has since traversed through CNN, culminating in the contemporary adoption of transformer architectures. This iterative refinement underscores the progressive evolution of EEG models, with each successive iteration enhancing the efficacy and sophistication of emotion recognition systems.

The leading model within this domain remains the refined EEG emotion recognition model grounded in transformer frameworks. However, recent investigations indicate that simultaneous utilization of multiple models yields superior outcomes. Consequently, it becomes evident that innovation gravitates towards transformer-based models as the predominant direction within the EEG-based emotion recognition domain. Concurrently, an increasing number of researchers are embarking on explorations involving integrating various models, a trend poised to gain further traction in the foreseeable future.

**Table 3   Summary table of subject-dependent articles.**

| Method | References | Dataset | Accuracy | | | |
|---|---|---|---|---|---|---|
| | | | DEAP | | SEED | SEED-IV |
| | | | Arousal | Valence | | |
| LSTM | *Garg et al. (2019)* | DEAP | 83.86% | 84.89% | – | – |
| LSTM+RNN | *Xing et al. (2019)* | DEAP | 74.38% | 81.10% | – | – |
| LSTM | *Li et al. (2019b)* | SEED | – | – | 93.38% | – |
| LSTM | *Acharya et al. (2020)* | SEED | – | – | 89.34% | – |
| LSTM | *Du et al. (2020)* | DEAP+SEED | 72.97% | 69.06% | 90.92% | – |
| RNN | *Prawira et al. (2021)* | SEED | – | – | 80.06% | – |
| LSTM | *Abgeena & Garg (2023)* | SEED | – | – | 97.83% | – |
| CNN+RNN | *Yang et al. (2018)* | DEAP | 91.03% | 90.80% | – | – |
| CNN | *Yang, Han & Min (2019)* | DEAP | 90.65% | 90.01% | – | – |
| CNN | *Hwang et al. (2020)* | SEED | – | – | 90.41% | – |
| CNN | *Chao & Dong (2020)* | DEAP | 97.34% | 96.46% | – | – |
| CNN | *Wang et al. (2020)* | DEAP+SEED | 82.84% | 82.84% | 90.59% | – |
| CNN | *Li et al. (2021a)* | DEAP | 94.59% | 95.32% | – | – |
| CNN | *Song et al. (2021)* | SEED | – | – | 92.93% | – |
| CNN | *Li et al. (2021b)* | DEAP | 79.30% | 75.90% | – | – |
| CNN | *Xiao et al. (2022)* | DEAP+SEED +SEED-IV | 97.39% | 96.90% | 96.25% | 86.77% |
| CNN | *Li et al. (2022a)* | DEAP | 98.51% | 98.35% | – | – |
| CNN | *Sun et al. (2022a)* | DEAP+SEED | 98.53% | 98.27% | 97.64% | – |
| CNN | *Bao et al. (2022)* | SEED+SEED-IV | – | – | 95.08% | 85.52% |
| CNN+LSTM | *Zhong et al. (2023a)* | SEED | – | – | 95.33% | – |
| CNN | *Li et al. (2023b)* | DEAP | 96.89% | 96.79% | – | – |
| CNN+LSTM | *Ramzan & Dawn (2023)* | DEAP+SEED | 97.81% | 97.77% | 93.74% | – |
| CNN+LSTM | *Iyer et al. (2023)* | SEED | – | – | 97.16% | – |
| CNN | *Aldawsari, Al-Ahmadi & Muhammad (2023)* | DEAP | 95.300% | 95.30% | 97.60% | – |
| CNN | *Yuvaraj et al. (2023)* | SEED | – | – | 90.85% | 83.71% |
| CNN | *Dondup, Manikandan & Cenkeramaddi (2023)* | SEED | – | – | 90.33% | – |
| Transformer | *Arjun, Rajpoot & Panicker (2021)* | DEAP | 99.10% | 99.40% | – | – |
| Transformer | *Wang et al. (2021b)* | DEAP+SEED | 98.55% | 98.51% | 97.30% | – |
| Transformer | *Aadam et al. (2022)* | DEAP | 91.49% | 90.41% | – | – |
| Transformer | *Li et al. (2022b)* | DEAP | 98.66% | 98.68% | – | – |
| Transformer | *Sun et al. (2022b)* | DEAP+SEED +SEED-IV | 94.61% | 95.91% | 97.31% | 89.97% |
| Transformer | *Liu, Zhou & Zhang (2022)* | DEAP | 97.02% | 96.76% | – | – |
| Transformer | *Song et al. (2022)* | SEED | – | – | 95.30% | – |
| Transformer | *Guo et al. (2022)* | SEED | – | – | 93.83% | – |
| Transformer | *Xu et al. (2023b)* | DEAP+SEED +SEED-IV | 97.48% | 96.85% | 97.17% | 87.32% |
| CNN+ Transformer | *Gong et al. (2023)* | SEED+SEED-IV | – | – | 98.47% | 91.90% |
| Transformer | *Lu, Tan & Ma (2023)* | SEED+SEED-IV | – | – | 97.55% | 88.08% |
| Transformer | *Qian et al. (2023)* | SEED | – | – | 92.76% | – |
| Transformer | *Sun et al. (2023)* | DEAP | 96.80% | 96.00% | – | – |

**Table 3** (*continued*)

| Method | References | Dataset | Accuracy | | | |
|---|---|---|---|---|---|---|
| | | | DEAP | | SEED | SEED-IV |
| | | | Arousal | Valence | | |
| Transformer | *Cao et al. (2023)* | DEAP | 95.15% | 94.41% | – | – |
| CNN+ Transformer | *Wan et al. (2023)* | SEED | – | – | 91.58% | – |
| Transformer | *Zhong et al. (2023b)* | DEAP | 96.63% | 96.69% | – | – |
| Transformer | *Wei et al. (2023)* | DEAP | 98.81% | 98.76% | – | – |
| Transformer | *Cai et al. (2023)* | DEAP | 99.05% | 99.05% | – | – |
| Transformer | *Lu, Ma & Tan (2023)* | SEED+SEED-IV | – | – | 98.57% | 92.09% |

## Subject-independent

With the ongoing advancements in human–computer interaction and brain-computer interfaces, a growing body of research focuses on EEG signals. Increasingly, researchers are turning their attention to the development of subject-independent emotion recognition algorithms. This research direction poses more significant challenges compared to its predecessor. The challenges stem from inherent individual differences, including variations in physiological aspects such as brain structure, head size, and unique emotional responses to the same stimuli. Consequently, subject-independent research becomes particularly intricate due to these inter-individual differences.

Nonetheless, the demand for large-scale datasets and contemporary applications necessitates developing models independent of individual idiosyncrasies. This need is increasingly observed in diverse fields, including large-scale studies in sociology and psychology, initial diagnostics of mental health disorders, and the implementation of intelligent recommender systems. These applications rely heavily on the model's generalization capability, enabling it to operate independently of individual variations.

In the design of neural networks and models for subject-independent research, a key distinction is observed in dataset partitioning. Unlike the subject-dependent approach, where datasets are divided collectively, in the subject-independent paradigm, datasets are segregated for individual participants. In this approach, the training and testing sets are distinctly associated with different individuals, framing the problem as a solution to the Source Data and Target Data Mismatch issue. This mismatch arises due to the substantial individual variability inherent in EEG signals. In essence, the subject-independent direction strives to address the inconsistency between the training and testing data, seeking to extract a limited set of standard features rather than individualized characteristics. Therefore, this research direction presents a formidable challenge compared to the preceding one.

In recent years, there has been a growing trend among researchers to integrate domain adversarial and domain adaptation models into subject-independent emotion recognition algorithms. In various fields, these methodologies are frequently regarded as interchangeable due to their shared characteristics and mutual objective of facilitating the model's acquisition of domain-agnostic feature representations. Domain adaptation primarily centers on mitigating distributional variance, whereas domain adversarial methods employ adversarial training techniques. Within the domain of EEG emotion

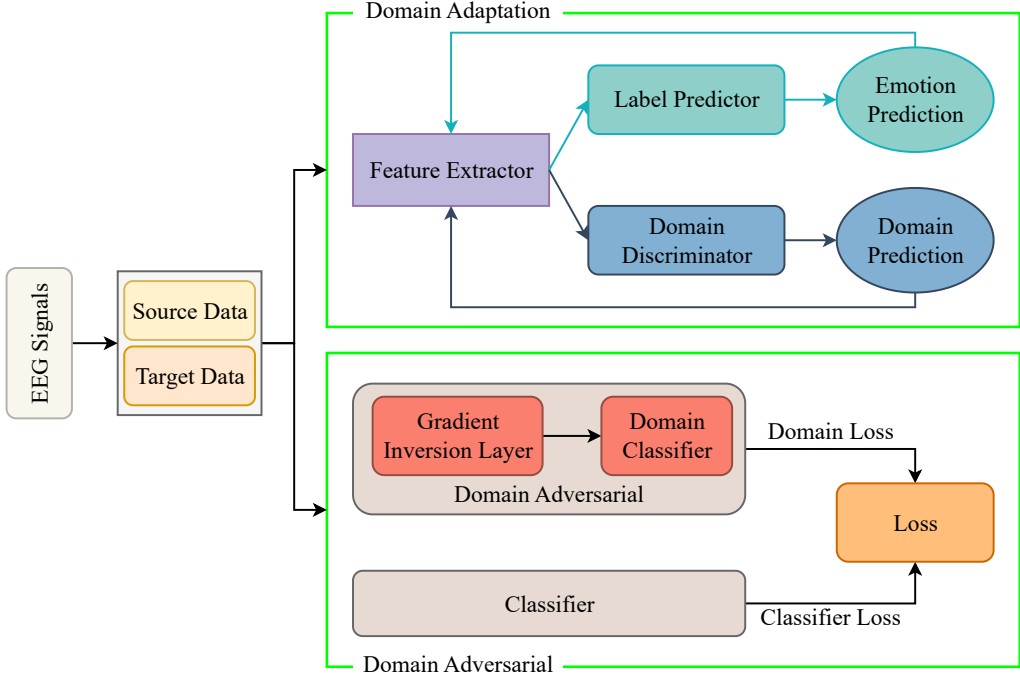

**Figure 5** Subject-independent technology roadmap.

recognition, these approaches may be delineated as distinct yet interconnected strategies. Figure 5 succinctly delineates the fundamental principles underlying these two avenues of research.

### Domain adversarial research

In the domain of domain adversarial research, *Li et al. (2018c)* proposes mapping EEG data from the right and left cerebral hemispheres separately into a discriminant feature space to aid in the classification of data representations. *Li et al. (2018d)* introduced a novel model known as the Bi-Hemisphere Domain Adversarial Neural Network. *Ma et al. (2019)* adapts deep adversarial networks for domain generalization and presents DResNet, an adversarial domain generalization framework. *Lew et al. (2020)* introduces RODAN, a regional operational domain adversarial network designed to capture the spatio-temporal relationships between brain regions and time. *Zhong, Wang & Miao (2020)* introduces two regularizers, NodeDAT and EmotionDL, to address cross-subject EEG changes and noisy labels, respectively, in emotion recognition tasks. *Wang et al. (2021a)* proposed the Few-Label Adversarial Domain Adaptation (FLADA) model, demonstrating superior performance when dealing with cross-individual scenarios with limited labeled data. *Ye et al. (2021)* introduces ADAAM-ER, an adversarial domain adaptive approach with an attentional mechanism to mitigate individual differences. *He, Zhong & Pan (2022)* presented the Adversarial Discriminative-Temporal Convolutional Networks (AD-TCNs), aiming to improve distribution alignment. *Tian et al. (2022)* proposes a spatial filtering

matrix to reduce overfitting, utilizes the feature extraction network 3DLSTM-ConvNET, and includes an effective local domain discriminator.

Pei et al. (2023) devised a Two-Step Domain Adversarial Transfer Learning (TS-DATL) framework, which exhibited substantial performance improvements, particularly on the DEAP dataset. Furthermore, Xu et al. (2023a) introduced a Domain Adversarial Graph Attention Model (DAGAM) model. Sartipi & Cetin (2023) proposes a method that integrates a transformer and an adversarial discriminative domain adaptation to perform emotion recognition across different subjects. Li et al. (2023e) presents STCL, which integrates an MLP for spatial feature learning and a transformer encoder for temporal feature learning. Additionally, it employs an adversarial training strategy to address domain gaps among different subjects. Gu et al. (2023) introduces a hybrid model that uses a generative adversarial network to generate latent representations of EEG signals, combining graph CNN and extended short-term memory networks for emotion recognition. Liu et al. (2023) introduces ATAM, a multi-task framework that combines adversarial training and attention mechanism using EEG and eye movement signals. Jiang et al. (2023) proposes MS-ADRT, a multi-source attention-based dynamic residual transfer approach, achieving multi-source domain adaptation through adversarial training based on maximum mean difference and maximum classifier difference.

### Domain adaptation research

In contrast, in domain adaptation research, Li et al. (2018a) introduced the deep adaptation network. Lan et al. (2018) demonstrated the effectiveness of domain adaptation techniques in enhancing classification accuracy. Li et al. (2019a) proposes a domain adaptation approach to generalize emotion recognition models across subjects and time. It minimizes source classification errors while aligning source and target domain representations. Chen et al. (2021) proposes MS-MDA, a method that considers domain invariance and domain-specific features through Multi-source Marginal Distribution Adaptation. Li et al. (2021c) introduces an innovative domain adaptation architecture utilizing adversarial training. Its goal is to learn domain-invariant features that generalize across different subjects. Liu, Guo & Hu (2021) introduces JAN, a novel joint adaptive network designed to address EEG-based emotion recognition tasks. Bao et al. (2021) devised the two-level domain adaptation neural network (TDANN). Ning, Chen & Zhang (2021) introduces SDA-FSL, a single-source domain adaptive few-shot learning network for cross-subject. This marks the first application of a domain adaptive approach with few-shot learning in EEG emotion recognition. Zhao, Yan & Lu (2021) proposed a Plug-and-Play Domain Adaptation approach. Cui et al. (2022) introduces a novel GRU-MCC model, combining gated recurrent units with minimum class confusion (MCC) loss during training. Zhu et al. (2022) presents TAGAT, a graph attention-based method for spatio-temporal pattern learning. It incorporates a domain discriminator into a domain adaptation-based model for subject-independent tasks. Li et al. (2022e) presents HVFM, a novel model capturing flow information between horizontal and vertical transitions of EEG channels. It extracts invariant emotional features using a domain discriminator. Li et al. (2022f) presents a dynamic training strategy where the model initially optimizes full-domain variance in

early training steps, gradually transitioning to focus on local sub-domain variances. *Peng et al. (2022)* proposes JAGP, a model that integrates domain-invariant feature learning, emotional state estimation, and optimal graph learning into a single objective.

*Li et al. (2023f)* introduced the Multi-Source Feature Representation and Alignment Network to account for the individual variability inherent in EEG signals effectively. Additionally, *Li et al. (2023d)* presented a neural network that combines Transposition Multi-Layer Perceptron (TMLP) with Sample-Reweighted Domain Adaptation Neural Network (SRDANN) to enhance model robustness. *She et al. (2023)* unveiled a novel emotion recognition approach by utilizing a multi-source associated domain adaptation (DA) network, which considers both domain-invariant and domain-specific features. *Jiménez-Guarneros & Fuentes-Pineda (2023b)* formulated the Multi-Source Feature Alignment and Label Rectification (MFA-LR) model, which, at the time, outperformed other models on publicly available datasets. *Wang et al. (2022a)* introduces MMDA-VAE, a multimodal domain adaptive variational autoencoder method that learns a shared cross-domain latent representation of multimodal data. *Mirzaee et al. (2023)* proposes MDA-NF, a fusion method based on the Neuro-Fuzzy Inference System, to combine classifiers using fuzzy membership functions for maximizing category separation. *Jiménez-Guarneros & Fuentes-Pineda (2023a)* proposes an SSDA framework to align joint distributions of subjects, emphasizing the importance of aligning fine-grained structures for improved knowledge transfer. *Quan et al. (2023)* introduces a cross-subject EEG emotion recognition algorithm with multi-source domain selection and subdomain adaptation, utilizing a multi-representation variant autoencoder (MR-VAE) for feature extraction.

### Other models

In the subject-independent domain, alongside the extensively researched domain adversarial and domain adaptation approaches, alternative models have been explored to enhance subject-independent emotion recognition. These include, among others, deep neural network architectures leveraging CNNs and Transformers. Each of these methodologies brings unique insights and strategies to tackle the inherent challenges of subject-independent emotion recognition. Through an exhaustive exploration of these diverse approaches, researchers aim to unearth novel insights and forge resilient frameworks capable of surmounting domain-specific constraints.

*Song et al. (2018)* introduces a multi-channel EEG emotion recognition method utilizing a novel dynamic graph convolutional neural network (DGCNN). *Cimtay & Ekmekcioglu (2020)* employs a pre-trained CNN architecture and applies a median filter to remove false detections within the sentiment prediction interval. *Li et al. (2022c)* introduces a novel ensemble learning method utilizing multi-objective particle swarm optimization. *Wang et al. (2022b)* introduces a transformer-based model to hierarchically learn spatial information from electrode to brain area levels, enhancing spatially dependent EEG capture. Subject-independent experiments were conducted on the DEAP database. *Pusarla, Singh & Tripathi (2022)* introduces an efficient method for extracting and classifying emotion-related information from two-dimensional spectrograms derived from one-dimensional EEG signals. *Chang et al. (2022)* preprocesses data with filtering and Euclidean alignment,

extracts time-frequency features using short-time Fourier transform and Hilbert-Yellow transform, utilizes CNN for spatial feature extraction, and explores bi-directional LSTM for temporal relationships. _Bai et al. (2023)_ introduces SECT, a method to extract emotion representation information from global and local brain regions. _Li et al. (2023a)_ employed a transformer-based encoder to capture time-frequency characteristics from EEG data, then utilized a spatial-time graph attention mechanism for emotion recognition. _Xu, Guo & Wang (2023)_ introduces GRU-Conv, a hybrid deep learning framework combining the strengths of both GRU and CNN architectures. _Tian et al. (2023)_ proposes DEVAE-GAN, a dual-encoder variational autoencoder-generative adversarial network, to generate high-quality artificial samples by incorporating spatio-temporal features.

Table 4 provides a comprehensive summary of articles and models in the subject-independent direction over the past six years. Much research in this field focuses on domain adversarial and domain adaptation methodologies. This prevalence can be attributed to the central challenge within the subject-independent direction, which addresses inconsistencies between source and target data. However, it is noteworthy that alternative research directions exist wherein various models are employed synergistically, yielding improved results. These approaches highlight the potential for leveraging a combination of methodologies to tackle subject-independent emotion recognition more effectively.

## SIGNIFICANCE AND APPLICATIONS

As deep learning continues to advance and researchers delve deeper into the field, the accuracy of EEG-based emotion recognition algorithms has significantly improved. This progress has enabled us to make increasingly accurate emotional predictions based on EEG signals. Consequently, the current focus has shifted towards applying these established models in human–computer interaction and brain-computer interfaces. This chapter explores the prospective downstream tasks to which these existing models can be applied. It delves into the various domains in which EEG emotion recognition can be effectively harnessed and the practical significance of deploying these models.

One compelling application of these models is in the diagnosis and treatment of psychological disorders, including but not limited to depression, agoraphobia, and post-traumatic stress disorder (PTSD). By leveraging the existing models, we can extend their utility to assess relaxation levels, degrees of depression, and tension levels based on EEG signals. Given the inherent similarities between emotion recognition and these psychological states, we can employ transfer learning techniques using pre-existing models to diagnose various mental health conditions. Furthermore, the existing models can be fine-tuned to classify EEG signals with a higher degree of granularity, enabling us to gauge the severity of conditions like agoraphobia. With the ongoing advancements in virtual reality and augmented reality technologies, an increasing number of studies have been exploring the utilization of these technologies for disease intervention and treatment (_Pira et al., 2023_; _Hawajri, Lindberg & Suominen, 2023_; _Purwar & Singh, 2023_; _Park et al., 2019_). This opens up the possibility of combining EEG signals with virtual reality technology. By

**Table 4  Summary table of subject-independent articles (Note: "A": domain adversarial, "B": domain adaptation, "C": other).**

| Method | References | Dataset | Accuracy | | | |
|---|---|---|---|---|---|---|
| | | | DEAP | | SEED | SEED-IV |
| | | | Arousal | Valence | | |
| A | Li et al. (2018c) | SEED | – | – | 92.38% | – |
| A | Li et al. (2018d) | SEED | – | – | 84.14% | – |
| A | Ma et al. (2019) | SEED | – | – | 87.07 | – |
| A | Lew et al. (2020) | DEAP+SEED-IV | 56.60% | 56.78% | – | 60.75% |
| A | Zhong, Wang & Miao (2020) | SEED+SEED-IV | – | – | 85.30% | 73.84% |
| A | Wang et al. (2021a) | DEAP+SEED | 68.00% | 68.00% | 89.32% | – |
| A | Ye et al. (2021) | SEED | – | – | 86.58% | – |
| A | He, Zhong & Pan (2022) | DEAP | 63.25% | 64.33% | – | – |
| A | Tian et al. (2022) | SEED | – | – | 97.40% | – |
| A | Pei et al. (2023) | DEAP | 60.42% | 71.89% | – | – |
| A | Xu et al. (2023a) | SEED+SEED-IV | – | – | 92.59% | 80.74% |
| A | Sartipi & Cetin (2023) | DEAP | 64.00% | 61.00% | | |
| A | Li et al. (2023e) | DEAP+SEED | 63.68% | 60.51% | 80.62% | – |
| A | Gu et al. (2023) | DEAP+SEED | 94.42% | 94.87% | 83.84% | – |
| A | Liu et al. (2023) | SEED+SEED-IV | – | – | 94.80% | 91.60% |
| A | Jiang et al. (2023) | SEED+SEED-IV | – | – | 90.81% | 68.98% |
| B | Li et al. (2018a) | SEED+SEED-IV | – | – | 83.81% | 58.87% |
| B | Lan et al. (2018) | DEAP+SEED | 48.93% | 48.93% | 72.47% | – |
| B | Li et al. (2019a) | DEAP+SEED | 62.66% | 62.66% | 88.28% | – |
| B | Chen et al. (2021) | SEED+SEED-IV | – | – | 89.63% | 59.34% |
| B | Li et al. (2021c) | SEED | – | – | 87.28% | – |
| B | Liu, Guo & Hu (2021) | SEED | – | – | 79.87% | – |
| B | Bao et al. (2021) | SEED | – | – | 87.89% | – |
| B | Ning, Chen & Zhang (2021) | DEAP+SEED | 67.62% | 67.62% | 97.66% | – |
| B | Zhao, Yan & Lu (2021) | SEED | – | – | 86.70% | – |
| B | Cui et al. (2022) | SEED | – | – | 88.07% | – |
| B | Zhu et al. (2022) | DEAP | 56.56% | 58.91% | – | – |
| B | Li et al. (2022e) | SEED | – | – | 92.75% | – |
| B | Li et al. (2022f) | SEED+SEED-IV | – | – | 91.08% | 81.58% |
| B | Peng et al. (2022) | SEED-IV | – | – | – | 78.77% |
| B | Li et al. (2023f) | DEAP+SEED | 57.03% | 59.31% | 82.02% | – |
| B | Li et al. (2023d) | DEAP+SEED | 57.70% | 61.88% | 81.04% | – |
| B | She et al. (2023) | DEAP+SEED +SEED-IV | 65.59% | 65.59% | 86.16% | 59.29% |
| B | Jiménez-Guarneros & Fuentes-Pineda (2023b) | SEED+SEED-IV | – | – | 89.11% | 74.99% |
| B | Wang et al. (2022a) | SEED+SEED-IV | – | – | 85.07% | 75.52% |
| B | Mirzaee et al. (2023) | DEAP | 68.50% | 72.00% | – | – |
| B | Jiménez-Guarneros & Fuentes-Pineda (2023a) | SEED+SEED-IV | – | – | 93.55% | 87.98% |
| B | Quan et al. (2023) | DEAP+SEED +SEED-IV | 79.59% | 81.19% | 92.83% | 79.30% |
| C | Song et al. (2018) | SEED | – | – | 79.95% | – |
| C | Cimtay & Ekmekcioglu (2020) | DEAP+SEED | 81.80% | 81.80% | 86.56% | – |

**Table 4** (*continued*)

| Method | References | Dataset | Accuracy | | | |
|---|---|---|---|---|---|---|
| | | | **DEAP** | | **SEED** | **SEED-IV** |
| | | | **Arousal** | **Valence** | | |
| C | *Li et al. (2022c)* | DEAP+SEED | 65.70% | 64.22% | 84.44% | – |
| C | *Wang et al. (2022b)* | DEAP | 66.20% | 66.63% | – | – |
| C | *Pusarla, Singh & Tripathi (2022)* | DEAP+SEED | 88.10% | 88.10% | 97.91% | – |
| C | *Chang et al. (2022)* | SEED | – | – | 86.42% | – |
| C | *Bai et al. (2023)* | DEAP+SEED | 65.31% | 66.83% | 85.43% | – |
| C | *Li et al. (2023a)* | SEED+SEED-IV | – | – | 90.37% | 76.43% |
| C | *Xu, Guo & Wang (2023)* | DEAP+SEED | 70.07% | 67.36% | 87.04% | – |
| C | *Tian et al. (2023)* | SEED | – | – | 97.21% | - |

utilizing existing models for patient diagnosis and employing virtual reality interventions, we can offer a more comprehensive approach to diagnosis and treatment. This synergy between EEG-based models and virtual reality technology has the potential to enhance the quality of care for individuals suffering from various mental health conditions.

EEG-based emotion recognition can pave the way for the development of recommendation systems, capitalizing on the inherent objectivity of EEG signals. As technology advances, there is a growing potential to create recommendation systems that rely on EEG-based insights. Unlike traditional recommendation systems that often use subjective methods such as questionnaires, EEG signals offer an objective and data-driven approach, enhancing the accuracy of recommendations. With the ongoing exploration of EEG signals, some recommendation systems have been developed based on this technology (*Guo & Elgendi, 2013*; *Panda et al., 2023*). These systems can be seamlessly integrated into psychotherapy applications, especially with virtual reality technology. Virtual scenes can be presented to patients in such scenarios for psychotherapeutic purposes. Simultaneously, these systems can utilize EEG signals to provide highly accurate recommendations. It is worth noting that only some virtual scenes may be suitable for some patients. Thus, the ability to make recommendations based on the patient's EEG signals, and even considering their specific mental health condition, becomes invaluable. By doing so, these systems can suggest virtual scenes that are better aligned with the patient's therapeutic needs, enhancing the overall effectiveness of the treatment process.

Classification models for emotion recognition using EEG signals hold significant relevance and can be applied to various downstream tasks and integrated with diverse domains. The continued research and development of these models is essential. Some researchers should focus on improving and extending existing models, pushing the boundaries of accuracy and applicability. Additionally, others should explore opportunities to integrate these models with different domains, maximizing their significance and utility in various practical applications. This collaborative effort will contribute to advancing EEG-based emotion recognition and its broader impact on different fields.

# DISCUSSION

Significant strides have been achieved in EEG-based emotion recognition, yet the field continues to harbor untapped potential and confronts various challenges necessitating further exploration. These challenges span multiple dimensions:

- **Challenges in datasets:** Existing datasets, constrained by technological limitations and the complexity of eliciting emotions, predominantly offer simplistic classifications ranging from three to four categories. Such categorizations inadequately capture the nuanced spectrum of human emotions. There is a pressing need for more granular classifications to enrich subsequent research endeavors. By augmenting datasets to include a broader array of emotional states, such as fear of heights or varying levels of relaxation, tailored data collection strategies, possibly incorporating psychotherapeutic methodologies, can significantly enhance downstream task performance. As technologies like Virtual Reality (VR) evolve, leveraging VR-induced emotions can enrich dataset collection efforts, bridging the gap between simulated and real-world emotional experiences.

- **Challenges in subject-dependent direction:** Despite considerable progress and commendable outcomes in this domain, opportunities for refinement persist. Future endeavors should prioritize enhancing the integration of temporal, spatial, and spectral features, particularly in bolstering feature extraction methodologies. Noteworthy is the burgeoning trend of amalgamating multiple models to compensate for individual model limitations, underscoring the importance of pursuing hybrid model architectures for feature extraction purposes.

- **Challenges in subject-independent direction:** While the research landscape increasingly gravitates toward subject-dependent investigations, the performance of models in subject-independent contexts still requires augmentation. Efforts should concentrate on fortifying model generalization and robustness. Encouragingly, recent undertakings have witnessed the emergence of novel model architectures and the exploration of model fusion techniques, offering promising avenues for bolstering the resilience and efficacy of subject-independent models.

- **Challenges in application:** Despite significant strides in model-centric research, the translational application of findings remains relatively underexplored. Future research trajectories should pivot towards application-oriented pursuits, extending beyond mere emotion recognition to encompass broader domains such as psychological diagnosis and treatment. EEG-based diagnostics hold considerable potential for revolutionizing psychological healthcare delivery, with the integration of VR technologies promising to amplify treatment efficacy, particularly for conditions where immersive interventions are advantageous. Moreover, the scope of application extends beyond conventional paradigms, necessitating sustained scholarly inquiry to unlock its full potential.

While significant progress has been made in EEG-based emotion recognition, multifaceted challenges persist across dataset development, modeling approaches, and

application domains. Addressing these challenges is critical to unlocking the field's full potential and ushering in emotion-centric research and applications.

## CONCLUSION

This article comprehensively reviews recent advancements in EEG-based emotion recognition techniques while introducing a systematic classification framework that categorizes predominant research into two primary streams: subject-dependent and subject-independent approaches. This framework is a valuable navigational aid for researchers, elucidating the intricate landscape of EEG-based emotion classification. Delineating the developmental trajectories and diverse avenues explored in this domain facilitates a deeper understanding of the foundational principles underlying these methodologies. Recognizing the substantial divergence in core principles and research priorities between subject-dependent and subject-independent approaches, this framework provides a clear delineation, aiding researchers in comprehending the nuanced aspects of existing methods and their application in respective directions. It elucidates each approach's specific objectives and techniques, offering clarity in a complex research domain.

Additionally, this article conducts a detailed examination of three established mainstream datasets alongside an emerging dataset, furnishing researchers with valuable insights into their characteristics and applications. This comparative analysis empowers researchers to make informed decisions regarding dataset selection, enhancing these invaluable resources' utility. Furthermore, the article delves into the applications and challenges within the field, providing valuable insights for subsequent researchers to grasp existing issues and chart future directions in the domain. By elucidating the practical applications and ongoing challenges, this article contributes to the collective understanding of EEG-based emotion classification, fostering continued progress and innovation in the field.

### Funding
This work was supported by the National Key R&D Program of China (NO. 2020YFC0811004). The funders had no role in study design, data collection and analysis, decision to publish, or preparation of the manuscript.

### Grant Disclosures
The following grant information was disclosed by the authors:
The National Key R&D Program of China: NO. 2020YFC0811004.

### Competing Interests
The authors declare there are no competing interests.

### Author Contributions
- Weizhi Ma conceived and designed the experiments, performed the experiments, analyzed the data, performed the computation work, prepared figures and/or tables, and approved the final draft.

- Yujia Zheng performed the experiments, prepared figures and/or tables, and approved the final draft.
- Tianhao Li performed the experiments, prepared figures and/or tables, and approved the final draft.
- Zhengping Li conceived and designed the experiments, authored or reviewed drafts of the article, and approved the final draft.
- Ying Li conceived and designed the experiments, authored or reviewed drafts of the article, and approved the final draft.
- Lijun Wang conceived and designed the experiments, authored or reviewed drafts of the article, and approved the final draft.

### Data Availability

This is a literature review.

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
