# Peer review of "A comprehensive review of deep learning in EEG-based emotion recognition: classifications, trends, and practical implications"

_PeerJ Computer Science, doi:10.7717/peerj-cs.2065_

## Round 0.1 · original submission · Major Revisions

Most of the reviewers have given critical comments on your paper. Their main concern was that the paper did not follow the required standard to meet the expectations of the review paper. So, I suggest the authors consider all the reviewer's comments carefully and substantially improve the quality of the manuscript to improve its standard.

**Language Note:** The review process has identified that the English language must be improved. PeerJ can provide language editing services - please contact us at [email protected] for pricing (be sure to provide your manuscript number and title). Alternatively, you should make your own arrangements to improve the language quality and provide details in your response letter. – PeerJ Staff

·

Basic reporting

This article is well written and easy to understand. Its structure is appropriate and it contains the sources expected for the topic. As a literature review, it fits in with the scope of the journal and covers an existing gap: The background of the topic as well as the existing literature (including reviews) is named and it is pointed out what the current gap is.

Experimental design

The Survey Methology chapter describes the procedure for selecting the relevant sources. However, the exact meaning of the criteria "demonstrating high research value and innovation" and how they were operationalized are not discussed here. It is suggested that this point be explained in more detail before publication.

Validity of the findings

The procedure for summarizing the original literature is comprehensible and plausible.

Reviewer 2 ·

Basic reporting

The authors provided a comprehensive overview of recent advances in EEG-based emotion classification techniques and proposed a systematic classification framework for subject-dependent and subject-independent approaches. The authors also provided a detailed analysis of datasets in the EEG-based emotion recognition area. The paper is well-written and readable. The quality of the article may further improve if the following queries are addressed.
1. The authors presented the literature review well, but it would be more beneficial for the readers if a summary table were given for such review papers.

Experimental design

2. Are there any specific reasons the authors did not consider papers from PubMed, and Web of Science platforms, as they also cover EEG-based emotion recognition papers?
3. The authors say that the sourced papers were screened initially. However, it is not clear what the metrics used for screening are.
4. The papers were sourced using a set of keywords and then classified. What are the specific keywords used (In Figure 1, it is said …, so it means that many more keywords than what is given in the paper were used.
5. Some of the details are not clear. For example, how many papers have been sourced using the keyword search for the last five years? After the screening, how many were selected? And under each classification category, how many papers were listed? This information helps the readers to know the recent research direction. Also, the discussion section may be strengthened using such details.

Validity of the findings

6. The discussion section is missing, and the conclusion section is lengthy.

Reviewer 3 ·

Basic reporting

EEG based emotion recognition is an extensively studied topic. First, it has started with basic ML and signal processing methods, and then followed by advanced signal processing methods as feature extraction. After DL, Convolutional, time-series, sequence modellings are studied using basic or processed inputs.

From this point of view, the road maps in this paper are not sufficient to be review paper. The recent and the most comprehensive apporaches in the field of DL are not included. Thus, it can not motivate the readers aboout the recent developments.

Finally, not suitalbe to be review paper.

Experimental design

Only, subject (in) dependent and cross-subject were written, but the combination of the past and the recent studies are not written. Thus, tecnically insufficient, especially classic ML and new aspects of the DL methods in this area.

Validity of the findings

in fact, I could not find valid things.

Reviewer 4 ·

Basic reporting

1. In the author's paper, mention of "physiological signals such as EEG, facial expressions, and speech signals" appears in line 39-42. However, it should be noted that facial expressions and voice signals are considered non-biological signals.
2. Lines 48-58 of the paper discuss the ability of convolutional neural networks and recurrent neural networks to enhance EEG features, and the potential impact on future research directions. While these networks are still commonly used, it is important to acknowledge the relevance of transformer networks. Recent literature should be consulted to demonstrate the importance of incorporating this method into future research endeavors.
3. The author introduces the content of a reference in lines 65-71, but fails to specify the prospective direction and primary findings of the cited material. Additionally, the paper overlooks the potential challenges and future research directions highlighted in lines 72-81.
4. The description of the DEAP dataset in lines 191-196 is inaccurate, as the baseline of the DEAP dataset is actually 3 seconds, not 5 seconds. Furthermore, in line 210, the author incorrectly states the position of Table 1.
5. Lastly, the manuscript contains numerous grammatical errors which could be rectified through a thorough proofreading by a fluent English language speaker.

Experimental design

no comment

Validity of the findings

no comment

Additional comments

no comment

---

## Round 0.2 · accepted · Accept

The author(s) have addressed the critical comments raised by the reviewer(s). The paper can be accepted for publication in PEERJ Computer Science.

·

Basic reporting

The authors of the paper addressed all of my remarks in the first review. Together with the modifications, the paper improved a lot. The literature review is now comprehensive.

Experimental design

The paper fits into the scope of the journal. The methods are described in sufficient detail.

Validity of the findings

The procedure for summarizing the original literature is comprehensible and plausible.

Reviewer 2 ·

Basic reporting

All previous queries are addressed well

Experimental design

All previous queries are addressed well

Validity of the findings

All previous queries are addressed well

Additional comments

All previous queries are addressed well

Reviewer 3 ·

Basic reporting

In revized version of the manuscript, deep learning methods with CNN, LSTM, and Transformers are sufficiently written and discussed. Roadmaps and related sections are now suitable.

Experimental design

subject (in) dependent,and related ML and DL approaches are well- organized.

Validity of the findings

The approaches on EEG emotion recognition are sequentially included, and discussed with their advantageous and disadvantageous.

Additional comments

Maybe, the year 2024 in fig1 can be removed.